# The Management and Outcomes of Patients with Extra-Pulmonary Neuroendocrine Neoplasms and Brain Metastases

Zainul-Abedin Kapacee [1], Jennifer Allison [1], Mohammed Dawod [1], Xin Wang [2], Melissa Frizziero [3], Bipasha Chakrabarty [4], Prakash Manoharan [5], Catherine McBain [6], Was Mansoor [1], Angela Lamarca [1], Richard Hubner [1], Juan W. Valle [1,7] and Mairéad G. McNamara [1,7,*]

1   Department of Medical Oncology, The Christie NHS Foundation Trust, Manchester M20 4BX, UK; zainul.kapacee@nhs.net (Z.-A.K.); j.allison@nhs.net (J.A.); mohammed.dawod@nhs.net (M.D.); was.mansoor@nhs.net (W.M.); angela.lamarca@nhs.net (A.L.); richard.hubner@nhs.net (R.H.); juan.valle@nhs.net (J.W.V.)
2   Statistics Group, Digital Services, The Christie NHS Foundation Trust, Manchester M20 4BX, UK; xin.wang@nhs.net
3   Cancer Research UK Manchester Institute, University of Manchester, Manchester M20 4BX, UK; melissa.frizziero@cruk.manchester.ac.uk
4   Department of Pathology, The Christie NHS Foundation Trust, Manchester M20 4BX, UK; bipasha.chakrabarty@nhs.net
5   Department of Nuclear Medicine/Radiology, The Christie NHS Foundation Trust, Manchester M20 4BX, UK; prakash.manoharan1@nhs.net
6   Department of Clinical Oncology, The Christie NHS Foundation Trust, Manchester M20 4BX, UK; catherine.mcbain1@nhs.net
7   Division of Cancer Sciences, University of Manchester, Manchester M13 9PL, UK
*   Correspondence: mairead.mcnamara@nhs.net

**Abstract:** Background: Brain metastases (BMs) in patients with extra-pulmonary neuroendocrine neoplasms (EP–NENs) are rare, and limited clinical information is available. The aim of this study was to detail the clinicopathological features, management and outcomes in patients with EP–NENs who developed BMs. Methods: A retrospective single-centre analysis of consecutive patients with EP–NENs (August 2004–February 2020) was conducted. Median overall survival (OS)/survival from BMs diagnosis was estimated (Kaplan–Meier). Results: Of 730 patients, 17 (1.9%) had BMs, median age 61 years (range 15–77); 8 (53%) male, unknown primary NEN site: 40%. Patients with BMs had grade 3 (G3) EP–NENs 11 (73%), G2: 3 (20%), G1: 1 (7%). Eight (53%) had poorly differentiated NENs, 6 were well-differentiated and 1 was not recorded. Additionally, 2 (13%) patients had synchronous BMs at diagnosis, whilst 13 (87%) developed BMs metachronously. The relative risk of developing BMs was 7.48 in patients with G3 disease vs. G1 + G2 disease ($p$ = 0.0001). Median time to the development of BMs after NEN diagnosis: 15.9 months (range 2.5–139.5). Five patients had a solitary BM, 12 had multiple BMs. Treatment of BMs were surgery ($n$ = 3); radiotherapy ($n$ = 5); 4: whole brain radiotherapy, 1: conformal radiotherapy (orbit). Nine (53%) had best supportive care. Median OS from NEN diagnosis was 23.6 months [95% CI 15.2–31.3]; median time to death from BMs diagnosis was 3.0 months [95% CI 0.0–8.3]. Conclusion: BMs in patients with EP–NENs are rare and of increased risk in G3 vs. G1 + G2 EP–NENs. Survival outcomes are poor, and a greater understanding is needed to improve therapeutic outcomes.

**Keywords:** brain metastases; extra pulmonary neuroendocrine neoplasms; neuroendocrine carcinoma

## 1. Introduction

Neuroendocrine neoplasms (NENs) arise from many different primary sites, with common sites being the gastroenteropancreatic system and lung [1]. Since the most recent

World Health Organisation (WHO) classification of NENs, subgroups can be broadly split into well-differentiated neuroendocrine tumours (NETs) or poorly differentiated neuroendocrine carcinomas (NECs) based on morphology, and they can be classified further into grade 1, 2 or 3 according to the Ki-67 labelling indexes on histological staining, which are <3%, 3–20% or >20% respectively [2].

Significant advances have been made in the therapeutic options for patients with grades 1 and 2 NETs, enabling improved survival and quality of life outcomes, even in the context of advanced disease [1]. By contrast, grade 3 NECs share similar biological properties to small cell lung carcinomas with a more aggressive phenotype, and for these patients, the survival outcomes are poor [3,4]. Some studies suggest that NECs may be sub-divided into two prognostically distinct entities, using cut-off Ki-67 indexes of >20 to <55% and ≥55%, with those of the higher Ki67 index tending to have a poorer prognosis [5,6]. As yet, there is no consensus on the optimum management strategy to treat grade 3 NECs, with most clinicians opting for platinum–etoposide combination regimens [7].

The incidence of NENs of non-lung origin or extra pulmonary neuroendocrine neo-plasms (EP–NENs) has increased in recent decades [8]. Patients with EP–NENs often present with advanced disease, and frequent sites of metastases include lymph nodes, liver and bone [9]. Brain metastases (BMs) in patients with EP–NENs are particularly rare; in a Surveillance, Epidemiology and End Results (SEER) registry of 14,685 patients with gastrointestinal NENs, of which 2003 patients had confirmed advanced disease, 27 patients (1.35%) developed BMs [10]. Another large cohort study of the Netherlands cancer reg-istry, which included 11,120 patients, observed BMs in 7/4710 (0.1%) and 4/1150 (0.3%) of patients with gastroenteral and pancreatic NENs, respectively [11]. A detailed sub cohort analysis of 539 patients from the same study found that 16% of patients with metastatic large cell pulmonary NEC had BMs at presentation compared with no patients having BMs at presentation with gastroenteral or pancreatic NENs, highlighting the different metastatic patterns between pulmonary and EP–NENs and the rarity with which BMs oc-cur in patients with EP–NENs. The incidence of BMs in other non-neuroendocrine cancers of the gastroenteropancreatic tract is comparable with that of EP–NENs; for example, in one systematic review of patients with metastatic colorectal cancer, 0.6–3.2% of patients developed BMs [12].

There is limited knowledge of the presentation, treatment options and outcomes of patients with EP–NENs and BMs, with no consensus guidelines on their management. In this retrospective study, at a European Neuroendocrine Tumour Society Centre of Excellence (ENETS CoE), the clinicopathological features, management and outcomes of patients with EP–NENs who developed BMs are described.

## 2. Materials and Methods

### 2.1. Study Design

A retrospective analysis of consecutive patients with EP–NENs managed at a ENETS CoE between August 2004 and February 2020 was performed (see Figure 1).

Electronic case records were reviewed to record baseline patient characteristics at diagnosis; tumour characteristics (grade, differentiation, proliferation index using Ki67%); disease site and stage; number of lines of systemic treatment and date of last clinical contact and death, where applicable.

Patients with EP–NENs who developed radiologically confirmed BMs on computed tomography (CT) or magnetic resonance imaging (MRI) were identified within this cohort. Specific details regarding the presenting symptomatology, number of BMs, site (cerebral, cerebellar, leptomeningeal, orbital) and management of the single or multiple BMs was recorded: surgery; whole brain radiotherapy (WBRT); conformal radiotherapy; or best supportive care (BSC), including use of corticosteroids. The time from date of initial diagnosis of EP–NEN to date of diagnosis of BMs was recorded in those with metachronous presentations. This study was approved by the internal review board of the Christie NHS Foundation Trust Clinical Audit Committee (reference 2746).

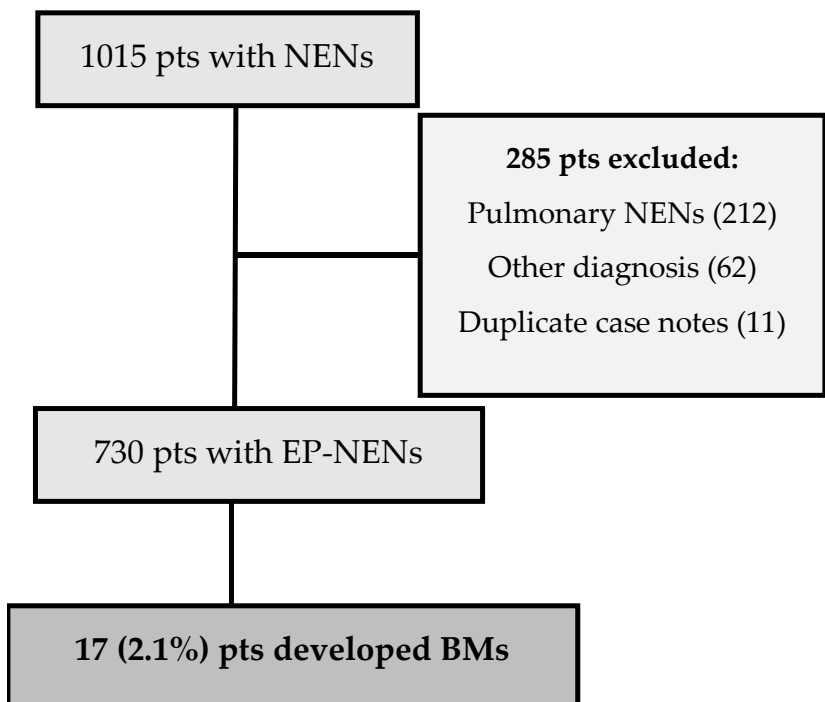

**Figure 1.** The CONSORT diagram of all new patient episodes with neuroendocrine neoplasm (NEN) diagnosis recorded and eligible patients remaining in the study following exclusions. Pts = patients. EP-NENs = extra pulmonary neuroendocrine neoplasms, BM=brain metastases.

### 2.2. Objectives

The aim of this study was to describe the incidence, clinicopathological characteristics, presentation, management and survival outcomes of patients with EP–NENs and BMs at an ENETS CoE. Survival outcomes were compared with patients with EP–NENs without BMs.

### 2.3. Statistical Analysis

Statistical analysis was performed using R software version 3.6.2. Patient demographics, clinicopathological characteristics and information on diagnostic and treatment modalities were presented as categorical variables in tables of frequency, except for patients' age, which was treated as a continuous variable and reported as median and range. Survival analyses were performed using the Kaplan–Meier method. Overall survival (OS) was estimated from date of histological diagnosis of EP–NEN to date of death, or date of last contact if alive. Survival from date of diagnosis of BMs to date of death or last contact was also estimated. Univariate and multivariate analyses were performed to identify potential prognostic variables for OS using the Cox proportional hazards model. Data were last updated 5 April 2021.

### 3. Results
#### 3.1. Patient Characteristics

Of 730 patients with EP–NENs, 17 patients (2.1%) developed BMs. The baseline characteristics of patients without BMs (*n* = 713) and those with BMs are summarised in Tables 1 and 2, respectively. Among patients who developed BMs, 10 (58.8%) were male and had a median age of 61 years (range 15–77; interquartile range 49–69). Fourteen (82%) patients had advanced disease at presentation, and the liver was the most common metastatic site (71.4%).

**Table 1.** Baseline characteristics of patients with extra-pulmonary neuroendocrine neoplasms without brain metastases (*n* = 713).

| Characteristics | Proportion *n* (%) |
|---|---|
| Gender | |
| Male | 397 (44.3) |
| Female | 316 (56.7) |
| Age, years, median (IQR) | 63 (52–71) |
| Baseline ECOG PS | |
| 0 | 246 (34.5) |
| 1 | 336 (47.1) |
| 2 | 91 (12.8) |
| 3 | 36 (5.1) |
| 4 | 4 (0.6) |
| Disease status at diagnosis | |
| Localised | 234 (32.8) |
| Advanced | 479 (67.2) |
| Primary site | |
| Small bowel | 292 (41.0) |
| Pancreas | 153 (21.5) |
| Large bowel | 72 (10.1) |
| Unknown primary | 71 (10.0) |
| Gastric | 46 (6.4) |
| Appendix | 31 (4.4) |
| Ovary | 7 (1.0) |
| Oesophagus | 6 (0.8) |
| Other | 28 (3.9) |
| Not recorded | 7 (1.0) |
| Histological grade | |
| 1 | 353 (49.5) |
| 2 | 178 (25.0) |
| 3 | 160 (22.4) |
| Not recorded | 22 (3.1) |
| Differentiation | |
| Well | 515 (72.0) |
| Moderate | 29 (4.2) |
| Poor | 148 (20.8) |
| Not recorded | 21 (3.0) |
| Ki67% | |
| $\leq 2$ | 297 (41.7) |
| 3-<20 | 174 (24.4) |
| $\geq$20-<55 | 58 (8.0) |
| $\geq$55 | 99 (13.9) |
| Not recorded | 85 (12.0) |

Abbreviations: IQR = interquartile range; ECOG PS= Eastern co-operative oncology group performance status.

The prevalence of BMs across the entire cohort (*n* = 730) by histological grade were 0.2%, 2.2% and 7.0% for grade 1, 2 and 3 NENs, respectively. In patients who developed BMs, a grade 3 NEN diagnosis (70.5%) and an unknown primary origin (41.2%) were predominant (Table 2). The relative risk of developing BMs increased with histological grade, with a relative risk of 7.48 (*p* = 0.0001) for developing BMs in patients with grade 3 disease compared with patients with grades 1 and 2 disease (see Table 3).

**Table 2.** Baseline characteristics of patients with extra-pulmonary neuroendocrine neoplasms and brain metastases (*n* = 17).

| Characteristics | Proportion *n* (%) |
|---|---|
| Gender | |
| Male | 10 (58.8) |
| Female | 7 (41.2) |
| Age, years, median (IQR) | 61 (49–69) |
| Baseline ECOG PS | |
| 0 | 7 (41.2) |
| 1 | 10 (58.8) |
| Disease status at diagnosis | |
| Localised | 3 (17.7) |
| Advanced | 14 (82.3) |
| Primary site | |
| Unknown primary | 7 (41.2) |
| Pancreas | 3 (17.6) |
| Oesophagus | 3 (17.6) |
| Small bowel | 2 (11.8) |
| Other | 2 (11.8) |
| Histological grade | |
| 1 | 1 (5.9) |
| 2 | 4 (23.5) |
| 3 | 12 (70.6) |
| Differentiation | |
| Well | 8 (47.1) |
| Poor | 8 (47.1) |
| Not recorded | 1 (5.8) |
| Ki67% | |
| $\leq 2$ | 1 (5.9) |
| 3–<20 | 4 (23.5) |
| $\geq$20–<55 | 4 (23.5) |
| $\geq$55 | 7 (41.2) |
| Not recorded | 1 (5.9) |
| Sites of other metastases | |
| Liver | 10 (71.4) |
| Bone | 7 (50.0) |
| Distant nodal | 6 (42.9) |
| Lung | 6 (42.9) |
| Lines of systemic therapy (*n*) | |
| 1 | 8 (47.1) |
| 2 | 3 (17.6) |
| 3 | 3 (17.6) |
| 4 | 3 (17.6) |

Abbreviations: IQR = interquartile range; ECOG PS = Eastern co-operative oncology group performance status.

From the sample, 15 patients who developed BMs metachronously had received prior systemic anticancer therapy; 7 (47.1%) had received a single line of treatment, 8 (52.9%) had received more than one line of treatment with 3, 3 and 2 patients had received 2, 3 and 4 lines of systemic treatment respectively. Systemic treatments were provided as directed by histological grading, patient performance status and in accordance with standard clinical practice [1].

**Table 3.** The relative risk ratios of developing brain metastases when comparing between histological grades of extra-pulmonary neuroendocrine neoplasms.

| | Patients with BMs | Patients without BMs | | Relative Risk | 95% CI | *p* Value |
|---|---|---|---|---|---|---|
| Grade 1 | 1 | 353 | G2 vs. G1 | 7.78 | 0.87–69.10 | 0.065 |
| | | | G3 vs. G1 | 24.68 | 3.24–188.40 | 0.002 |
| Grade 2 | 4 | 178 | G3 vs. G2 | 3.17 | 1.04–9.65 | 0.041 |
| Grade 3 | 12 | 160 | G3 vs. G1 + G2 | 7.48 | 2.67–20.93 | 0.0001 |

Abbreviations: CI = confidence interval, BMs = brain metastases. G = grade.

### 3.2. Presentation and Investigation of Patients with BMs

Two patients had synchronous BMs identified at initial diagnosis, whereas in 15 patients, BMs developed metachronously. The median times to the development of BMs in patients with grade 2 (*n* = 4) and grade 3 NENs (*n* = 11) were 52.0 months (range 19.4–139.5) and 7.4 months (range 1.1–23.1), respectively. The most common presenting symptoms were weakness/gait disturbance (23.5%) and confusion (23.5%). Seizures were less common, affecting 2 (11.8%) patients (Table 4). Five (29.4%) patients developed a solitary BM, six (35.3%) patients developed 2–10 lesions and six patients (35.3%) developed more than 10 lesions. In the five patients with a solitary BM, one patient had grade 1 disease (choroidal deposit), two had grade 2 disease and the remaining two had grade 3 disease. In the 12 patients with multiple BMs, 2 patients had grade 2 disease (both of pancreatic origin) and 10 patients had grade 3 disease.

**Table 4.** Presentation, investigations and management summary of brain metastases in patients with extra-pulmonary neuroendocrine neoplasms (*n* = 17).

| | Proportion *n* (%) |
|---|---|
| **Development of BMs** | |
| Synchronous | 2 (11.8) |
| Metachronous | 15 (98.2) |
| **Presenting symptoms of BMs** | |
| Weakness/gait disturbance | 4 (23.5) |
| Confusion | 4 (23.5) |
| Aphasia | 3 (17.6) |
| Headaches | 3 (17.6) |
| Visual disturbances | 3 (17.6) |
| Seizures | 2 (11.8) |
| **Number of BMs** | |
| 1 | 5 (29.4) |
| 2–10 | 6 (35.3) |
| >10 | 6 (35.3) |
| **Intracranial site of BMs** | |
| Cerebral | 15 (88.2) |
| Cerebellar | 6 (35.3) |
| Leptomeningeal | 2 (11.8) |
| Orbital | 1 (5.9) |
| **Imaging modality for diagnosis** | |
| CT head only | 3 (17.6) |
| MRI brain only | 9 (53.0) |
| CT head and MRI brain | 5 (29.4) |

**Table 4.** *Cont.*

|  | Proportion *n* (%) |
|---|---|
| Definitive treatment of BMs |  |
| Best supportive care +/− steroids | 9 (53.0) |
| Whole brain radiotherapy | 4 (23.5) |
| Surgical resection | 3 (17.6) |
| Other | 1 (5.9) |
| Follow up imaging |  |
| None performed | 13 (76.5) |
| MRI brain | 4 (23.5) |

Abbreviations: BMs = brain metastases.

Cerebral metastases were most common (88.2% of patients), six (35.3%) had cerebellar metastases and two patients (11%) had leptomeningeal involvement. In patients who developed metachronous BMs (*n* = 15), 9 (60%) patients had progression of extracranial disease, whilst 6 (40%) had stable disease, determined by the most recent restaging CT thorax abdomen and pelvis scans performed at the time of BM diagnosis. Magnetic resonance imaging alone was used to diagnose BMs in 9 (53%) patients, MRI and CT head in 5 (29.4%) patients and CT alone in 3 patients. In the 3 patients diagnosed by CT alone, 2 had multiple BMs, and MRI imaging would not have altered management for palliation; the third patient was unable to tolerate an MRI scan. The MRI scan axial images for one patient with a solitary metastasis and one with multiple metastases are shown in Figure 2.

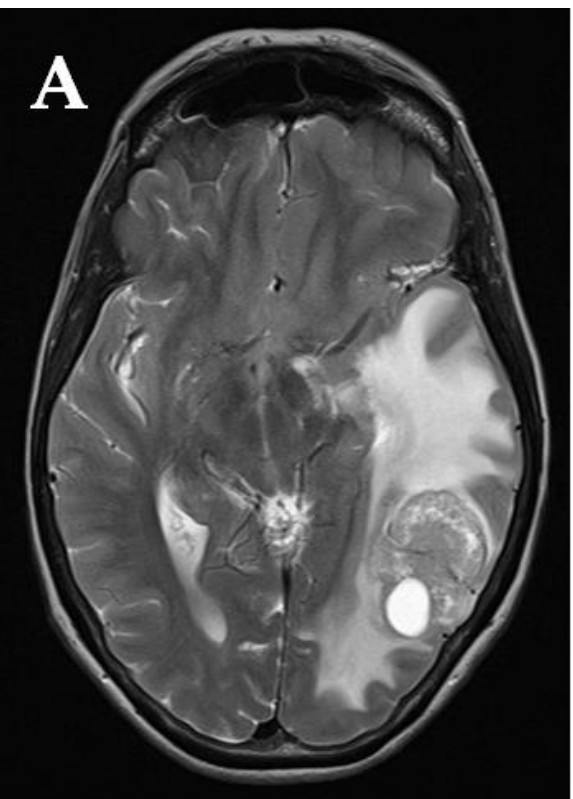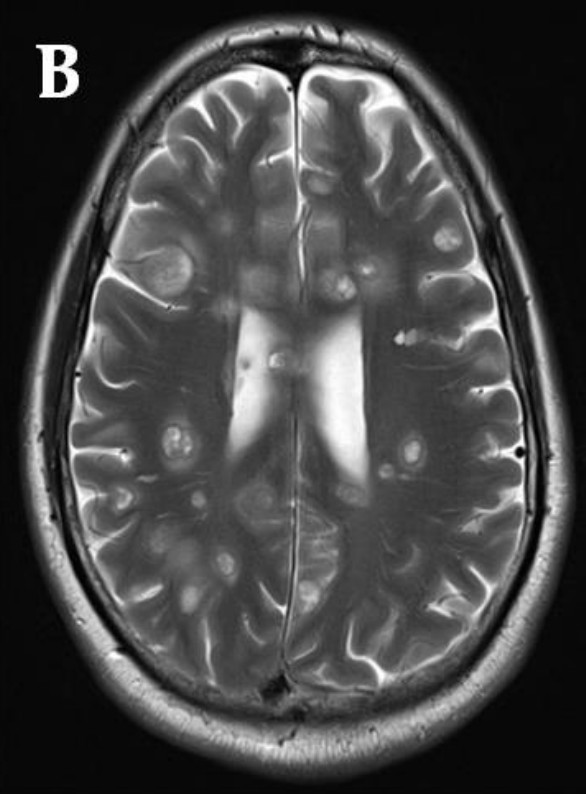

**Figure 2.** (**A**): T2 weighted magnetic resonance (MR) image of a solitary parietal lobe brain metastasis (BM) and vasogenic oedema; female, 67 years old, with grade 2 metastatic small bowel NEN. Patient was treated with high-dose steroids and underwent surgical resection of the solitary BM. (**B**): T2 weighted MR image of diffuse multiple BMs within cerebrum; male, 55 years old, with G3 metastatic NEC of unknown primary. Patient treated with high-dose steroids and best supportive care.

*3.3. Management of Patients with BMs*

Upon diagnosis of BMs, all patients received high-dose dexamethasone, with 9 patients (53%) receiving best supportive care (BSC) with steroids only. Treatment decisions for surgery and/or radiotherapy were made following discussion at neuro-oncology multidisciplinary team meetings that included the regional neurosurgical, radiation/clinical oncologist and neuro-radiology specialists. Surgical resection, whole-brain radiotherapy and conformal radiotherapy (to the orbit) were the other treatments given (see Table 4). In our cohort, no patients were deemed eligible to receive stereotactic radiosurgery (SRS). In the five patients with solitary BMs, 3 patients had surgical resection, one had conformal radiotherapy to the orbit (choroidal deposit) and one given BSC with steroids. In the 12 patients with multiple BMs, 4 received WBRT, and 8 were given BSC with steroids.

Of the 12 patients with grade 3 disease, 11 had a documented Ki67 index; 4 patients had a Ki67 index ranging 20–<55%, 1 of whom had surgical resection of a solitary metastasis; the other 3 received BSC. Another 7 patients had a Ki67 index >55%, 3 of whom received WBRT and 4 BSC.

### 3.3.1. Patients Who Underwent Surgical Resection of BMs

Three patients with a solitary BM underwent surgical resection, and all had heterogenous presenting and clinicopathological features. Two patients with grade 2 EP–NENs of small bowel origin developed solitary BMs metachronously and both had received two lines of systemic therapy (somatostatin analogue and temozolomide/capecitabine chemotherapy). One patient developed a frontal lobe metastasis that manifested with confusion and lethargy, whilst the other patient presented with word finding difficulty and developed a temporal lobe metastasis (see Figure 2). The resected frontal lobe metastasis had similar histopathological features to the primary disease of a grade 2 NEN of gastrointestinal origin, whilst in the patient with the resected temporal lobe metastasis, the histopathology demonstrated a grade 3 poorly differentiated NEC. After surgical resection, both patients developed intracranial disease recurrence after different time intervals. The patient with the frontal lobe metastasis developed diplopia with enlarging metastatic nodules in the left orbital muscles treated with palliative radiotherapy 20 Gray (Gy) in five fractions. At 15.9 months, this patient developed dural-based parietal metastases. The patient with the resected temporal lobe metastasis developed intracranial recurrence at 12.6 months.

The third patient was treated with definitive chemoradiotherapy for stage 1b high-grade NEC of the cervix and developed recurrent disease in the left breast after 86.4 months. An octreotide re-staging scan identified a solitary right occipital lobe metastasis that was histologically confirmed to be NEC. This patient also received prophylactic WBRT with 30 Gy in 10 fractions to reduce the risk of intracranial recurrence. This patient subsequently commenced systemic therapy with carboplatin and etoposide, but treatment toxicity and progressive clinical deterioration limited further treatment.

### 3.3.2. Patients Who Received Radiotherapy for BMs

Four patients received WBRT, and all had grade 3 poorly differentiated NECs. All had multiple BMs, were ineligible for stereotactic radiosurgery (SRS) and continued with dexamethasone steroid maintenance pre- and post-WBRT. The presenting symptom common to all four patients was confusion; two patients also had limb weakness, and one patient suffered seizures. One patient received 30 Gy in 10 fractions, two patients received 20 Gy in five fractions and one patient tolerated 4/5 fractions of a 20 Gy in a five-fraction regime. The clinical response to WBRT was varied: Two patients achieved stabilisation of neurological symptoms, but in two patients, no symptomatic benefit was gained with continued clinical deterioration.

One patient presented with blurred vision and was diagnosed with a left choroidal metastasis from a grade 1 NET of unknown primary. The patient received high-dose palliative radiotherapy with 40 Gy in 20 fractions to the left orbit, with improvement in visual symptoms reported at subsequent clinical reviews.

*3.4. Survival Outcomes*

In all patients with EP–NENs and BMs, the median OS was 16.8 months (95% CI 8.1–23.8), whilst in patients with advanced disease and without BMs, the median OS was 50.9 months (95% CI 39.0–61.7). Subdivided by low versus high grade, the median OS for patients with BMs and with grade 1 and 2 NENs (*n* = 5) was 73.0 months (95% CI 20.8-not estimable), whereas the median OS for patients with grade 3 NENs (*n* = 12) was 9.4 months (95% CI 7.8–16.8) (Figure 3). Comparatively, in patients with advanced disease without BMs, the median OS for patients with grade 1, 2 and 3 NENs was 95.8 months (95% CI 77.0–177.1), 61.7 months (95% CI 50.1–124.4) and 11.3 months (95% CI 9.3–14.4) respectively (Figure 4).

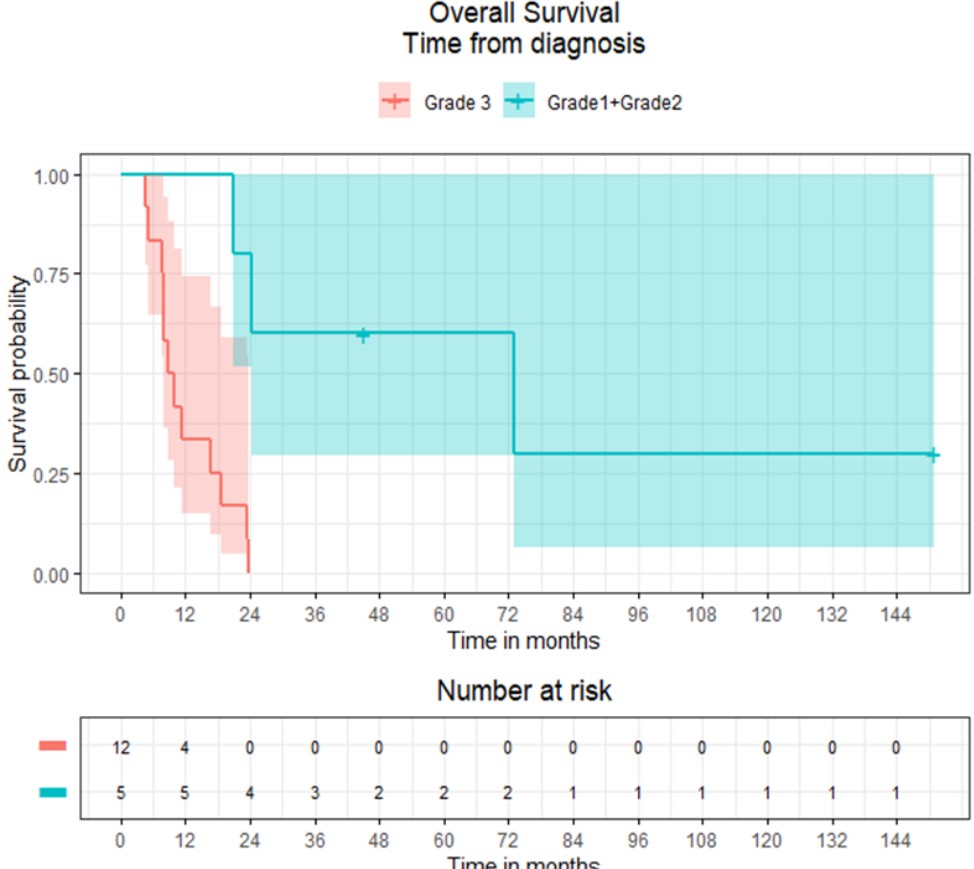

**Figure 3.** Kaplan–Meier overall survival curves of patients with extra-pulmonary neuroendocrine neoplasms and brain metastases by histological grades 1 + 2 (grouped) and 3.

The median survival from diagnosis of BMs in patients with histological grades 1 and 2 (grouped; *n* = 5) and grade 3 NENs (*n* = 12) was 24.2 months (95% CI 1.5, not estimable) and 2.4 months (95% CI 1.0–4.5) respectively (Figure 5). The estimated median survival times from BM diagnosis were 1.4, 6.2, 13.8 and 24.2 months in patients who received best supportive care, WBRT, surgery and conformal radiotherapy (orbit), but due to the small sample size, the 95% CI could not be accurately estimated. The histological grade (grades 1 + 2 vs. grade 3) and number of BMs (10 ≤ vs. >10) were statistically significant on univariate Cox regression analysis for OS in patients with EP–NENs, histological grade remained statistically significant on multivariate analysis (Table 5).

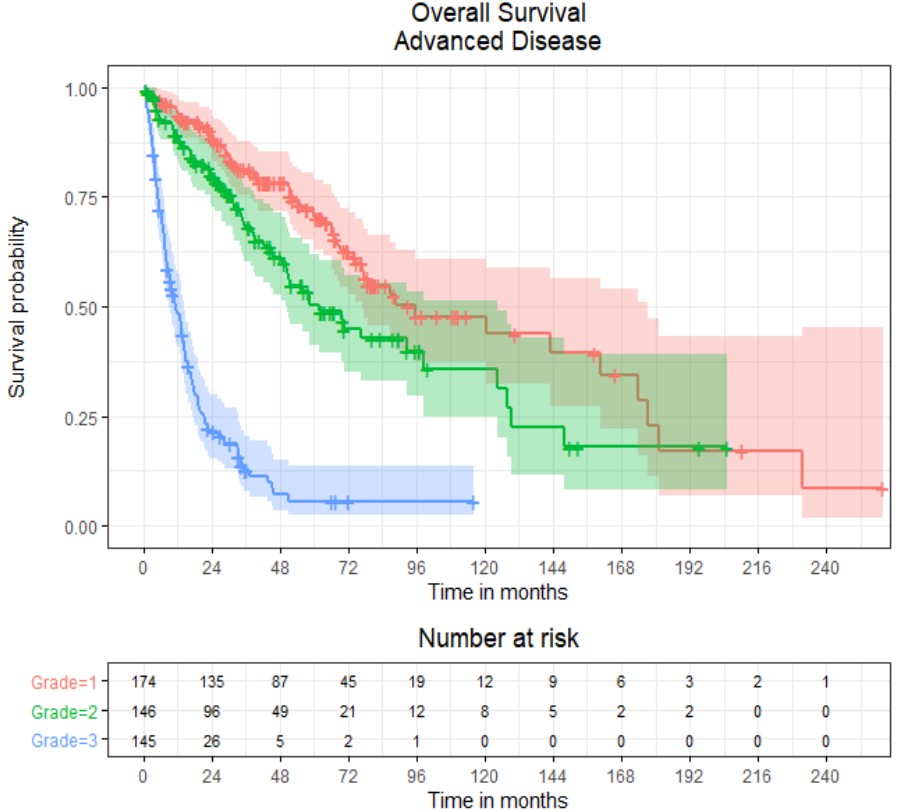

**Figure 4.** Kaplan–Meier overall survival curves of patients with advanced extra-pulmonary neuroendocrine neoplasms without brain metastases by histological grades 1, 2, and 3.

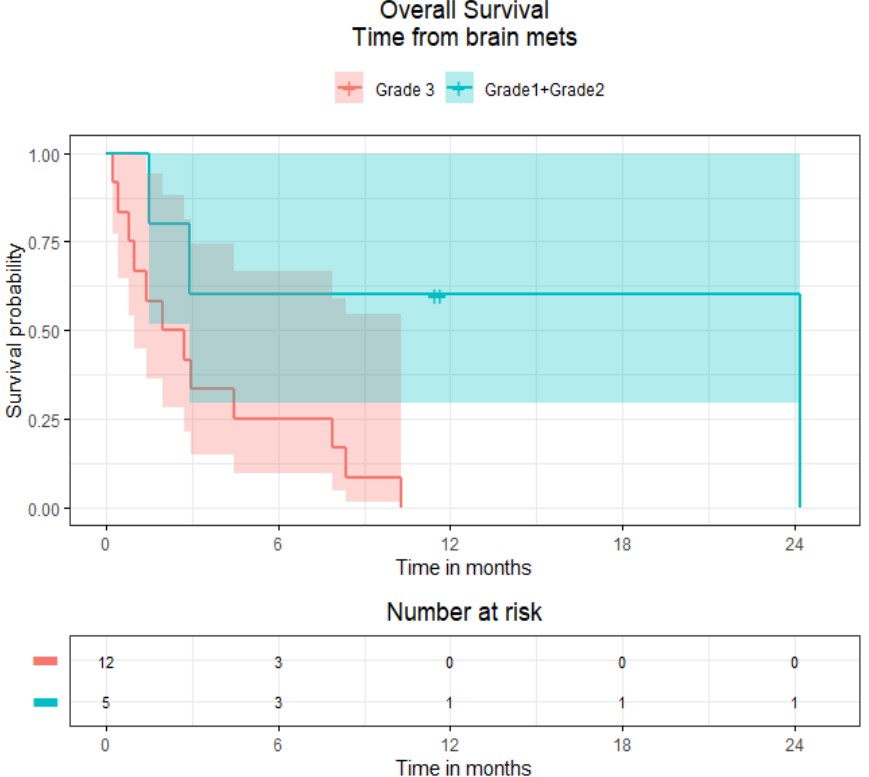

**Figure 5.** Kaplan–Meier survival curve from BM diagnosis to death for patients with grades 1 + 2 (grouped) versus 3 disease.

**Table 5.** Univariate and multivariate Cox regression analyses of factors associated with OS in patients with EP–NENs and BMs.

| Variable | HR | 95% CI | | *p* Value |
|---|---|---|---|---|
| | | Lower | Upper | |
| *Univariate* | | | | |
| Age | 1 | 0.97 | 1.03 | 0.89 |
| Sex | 1.55 | 0.56 | 4.30 | 0.40 |
| Tumour grade (Grade 1 = reference) | | | | |
| Grade 2 | 0.30 | 0.02 | 4.87 | 0.40 |
| Grade 3 | 6.6 | 0.52 | 84.12 | 0.14 |
| **Grade 3 vs Grade 1 + Grade 2** | **14.5** | **1.8** | **116.0** | **0.01** |
| Number of BMs (solitary = reference) | | | | |
| 2–10 BMs | 2.33 | 0.65 | 8.37 | 0.19 |
| **>10 BMs** | **5.95** | **1.29** | **27.55** | **0.02** |
| Treatment of BMs (BSC = reference) | | | | |
| Surgery | 0.28 | 0.06 | 1.34 | 0.11 |
| WBRT | 1.38 | 0.41 | 4.69 | 0.60 |
| Conformal radiotherapy (to orbit) | 0.45 | 0.05 | 3.80 | 0.47 |
| *Multivariate* | | | | |
| Grade 1 + 2 (referance) | | | | |
| **Grade 3** | **11.04** | **1.29** | **94.20** | **0.03** |
| Number of BMs (solitary = reference) | | | | |
| 2–10 BMs | 1.78 | 0.47 | 6.66 | 0.39 |
| >10 BMs | 2.32 | 0.49 | 10.94 | 0.29 |

Abbreviations: HR = hazard ratio, CI = confidence interval, WBRT = whole brain radiotherapy.

## 4. Discussion

This single tertiary-referral centre retrospective study provides an insight into the outcomes of patients with EP–NENs who develop BMs. This study demonstrates that BMs are uncommon in patients with EP–NENs but the relative risk increases with histological grade, and it adds to the existing minimal literature from retrospective data [10,13]. Only one patient with a grade 1 NET developed a presumed choroidal metastasis of the orbit, which further suggests that intracranial BMs in patients with grade 1 NENs are extremely rare. Patients with grade 2 EP–NENs developed BMs after a median time of 52.0 months, and 2 of the 4 patients had solitary BMs that were surgically resected. Interestingly, the histopathology of one of the patient's resected BMs was of a poorly differentiated NEC that was different to the primary grade 2 EP–NENs. This reflects the tumour heterogeneity, and it is likely that subpopulations from the primary tumour or other metastatic sites exist that possess unique mutations and properties to metastasise to the brain, with the potential ability to de-differentiate.

The relative risk of developing BMs was greatest in patients with grade 3 EP–NECs 7.48 (*p* = 0.0001), and the median time to the development of BMs was 7.4 months in this subgroup. Owing to the small number of patients, it was not possible to definitively report the risk of developing BMs based on Ki67 labelling index of 20 < 55% and ≥55%. On descriptive analysis, there were no apparent differences in the management of patients according to Ki67 index. On Cox regression analysis, grade 3 disease is associated with poorer overall survival compared with grades 1 + 2 disease, and this was found to be independent of the number of BMs detected on imaging. Whilst these findings demonstrate the aggressive nature of grade 3 NECs, these results should be interpreted with caution owing to the small number of patients and wide confidence intervals.

Our findings are similar to those from a retrospective study of 51 patients that found that grade 3 disease was an adverse prognostic factor in patients with gastroenteropancreatic and pulmonary NENs who then developed BMs [HR 4.2 (95% CI 1.1–16.1)]. Inter-

estingly, this study also found age over 60 to predict poorer survival outcome in patients with BMs, a finding not demonstrated in our study [13]. Another retrospective SEER study of 2005 patients that included pulmonary and EP–NENs found that patients with grades 3 and 4 (aplastic) disease had worse survival outcomes compared with patients with grades 1 and 2 disease HR 4.53 (3.54–5.79, *p* < 0.0001) [14]. This study also found the site of metastases to be prognostic for overall survival, with BMs associated with worse survival outcomes (HR 1.66 (95% CI 1.17–2.35 *p* < 0.006)) when compared with lung metastases. Both studies [13,14], however, include patients with pulmonary NENs, which had the greater prevalence of BMs; this might limit direct comparison with our study of patients with EP–NENs only.

Owing to the paucity of literature on the management of patients with NENs and BMs, comparisons are drawn from literature studying BMs in patients with small cell lung carcinoma (SCLC). Most patients with grade 3 NECs in our study developed multiple BMs, and the mainstay of treatment was best supportive care with steroids or, in select cases, WBRT. The rationale for WBRT is to achieve local tumour and symptom control and the potential tapering of high-dose corticosteroid use. From the descriptive analyses of the four patients who received WBRT, all were continued on dexamethasone, but in the absence of corticosteroid dosing information, it is difficult to determine if any potential benefits were due to WBRT or due to corticosteroids. In this current study, there was no statistically significant survival advantage for patients receiving WBRT compared with best supportive care. Although there is limited experience of WBRT in EP–NENs, BMs are more common in small cell and non-small cell lung cancers (NSCLC). One phase II trial of 22 patients with small cell lung cancer (SCLC) and multiple BMs reported a 50% response rate to 30 Gy in 10 fractions WBRT but a short median response duration of 5.7 months and OS of 4.3 months [15]. A phase III trial of 128 patients with SCLC and BMs randomised patients to receive WBRT and teniposide or teniposide only. This study found greater response rates with combined modality treatment compared with chemotherapy only; however, median OS was poor and no different in both groups at 3.5 and 3.2 months, respectively (*p* = 0.87) [16]. The QUARTZ clinical trial was a large multi-centre study comparing the survival and quality-of-life benefits of WBRT with BSC and steroids to BSC and steroids alone in patients with NSCLC and inoperable BMs. The primary outcome of this study was quality-adjusted life-years (QALYs) and were generated from overall survival and patients' weekly completion of a quality-of-life based questionnaire. This trial did not demonstrate a significant survival advantage or quality-of-life improvement in patients who received WBRT over BSC with a difference in QALYs of 4.7 days; 46.4 QALY days for the BSC plus WBRT cohort compared with 41.7 QALY days for the BSC group (hazard ratio (HR) 1.06, 95% CI 0.90–1.26) [17]. Prospective studies comparing neurocognitive and quality of life outcomes in patients receiving WBRT compared with other modalities such as stereotactic surgery (SRS) are underway [18]. Currently, the use of WBRT remains contentious, with no or modest improvements in survival outcomes or quality of life in other cancer sites, which is similar to the outcomes reported for the four patients who received WBRT in our study.

In our study, no patients were deemed eligible for SRS, which utilises high-dose conformal radiotherapy to the sites of metastases. Traditionally, SRS was recommended for patients with a maximum of three intracranial lesions and superior survival outcomes achieved when combined with WBRT [19]. In the UK, the Royal College of Radiology (RCR) guidelines advocate that total volume of irradiated intracranial disease is a more apt measure and that SRS can be offered for BMs of a maximum total volume of 20 cm$^3$ [20,21]. The American Society of Clinical Oncology (ASCO), the Society for Neuro-Oncology (SNO) and the American Society for Radiation Oncology (ASTRO) recommend SRS as a suitable therapy for patients with 1–4 BMs of less than 4 cm in size of solid tumour origin [22]. Crucially, however, this recommendation excludes patients with SCLCs owing to the paucity of data from randomised controlled trials. The evidence base for treating BMs with SRS in patients with SCLCs and NENs is largely limited to retrospective studies [23]. In one study of patients with large cell NEC of the lung, 348/9970 patients (3.4%) developed

BMs. In that cohort, 68 patients were treated with SRS and 280 with WBRT, with better survival outcomes for patients treated with SRS compared with WBRT, with a median OS of 11 months and 6 months, respectively ($p = 0.007$) [24]. Similarly, a Japanese study of 101 patients with large cell pulmonary NEC, with a median of 3 BMs treated with SRS, report a median OS of 9.6 months [25]. By contrast, a multicentre cohort study that analysed SRS outcomes to first-line WBRT for patients with SCLC demonstrated an improved time to progression of intracranial disease with WBRT compared with SRS (HR 0.38 95% CI 0.26–0.55, $p < 0.001$). However, no difference was observed in overall survival [26]. Patients undergoing SRS require adequate performance score (Karnofsky performance score > 70) and intact cognition to adequately consent and tolerate the treatment, and they must also have controlled extracranial disease. The majority of patients in our study had multiples BMs, and confusion was a common presenting symptom along with progression of extracranial disease, which were the likely factors precluding SRS.

Prophylactic cranial irradiation (PCI) is a management strategy utilized in treating patients with SCLC in both limited-stage and extensive disease. In limited-stage SCLC, post-chemoradiotherapy, the brain is a potential site of relapse, and in one meta-analysis, PCI demonstrated a 5.4% improvement in OS at 3 years versus no treatment (HR 0.84, 95% CI 0.73–0.97), with additional improvements in disease-free survival (relative risk (RR) 0.75, 95% CI 0.65–0.86) and risk of development of BMs (RR 0.46, 95% CI 0.38–0.57) [27]. In extensive-stage SCLC, PCI offered to selected patients can reduce the risk of developing symptomatic BMs (HR 0.27, 95% CI 0.16–0.44, $p < 0.001$) [28]. The incidence of BMs is significantly higher in SCLC by comparison, which can be in excess of 18% at diagnosis and 50% by 2 years of diagnosis providing potential justification for PCI [29]. There are few small retrospective studies of patients with extra-pulmonary small cell cancers who received PCI but did not find any improvement in outcomes [30,31]. In the context of EP–NENs, in which development of BMs are rare [10,11,13,14], there is a lack of evidence to determine if PCI would be of any benefit in these patients.

This study is limited by its retrospective nature and the small frequency of patients with BMs, but it contributes to the paucity of literature currently available. Much of the discussion into management strategies of BMs extrapolates from literature from small cell lung cancers, which share some biological similarities to EP–NECs, although the pathophysiology of developing BMs and the responses to therapies are likely to differ. Survival analyses must be interpreted with caution, and this study is unable to offer clinical validation for any potential prognostic factors. At present, it is not routine practice for brain imaging as part of staging or surveillance of patients with advanced EP–NENs, which may fail to capture patients with potentially asymptomatic BMs. There may also be an underestimation of patients with BMs, as only those patients who presented with neurological symptoms and had cranial imaging performed or imaging results in some way documented at our institution will have been captured in this study. Being a tertiary cancer centre, many patients, particularly those with grade 3 NECs, who exhibited rapid disease progression and deterioration may have died in other hospitals or in the community setting with undiagnosed BMs.

Based on this single-centre ENETS CoE experience, our suggestions are:

- Magnetic resonance imaging of the brain should be considered in patients with new neurological symptoms, particularly in patients with grade 3 NEC.
- Surgical resection for a solitary metastasis could be offered if the patient has well-differentiated disease, a good performance status, stable extra-cranial disease and/or further lines of treatment available. Stereotactic radiosurgery, with or without WBRT, may be a treatment option for low-volume metastatic disease, where surgical resection is not feasible.
- The role of WBRT alone is unclear and only a subset of patients may derive benefit

The prognosis for patients with EP–NENs and BMs is poor, and there is a void of evidence to develop consensus guidelines for the optimal management for these patients. Prospective studies or randomised control trials would unlikely be feasible in patients with

EP–NENs and BMs. Future studies of this cohort of patients may require collaboration amongst ENETs CoEs, using standardised methods to gain a multicentre insight and ultimately help devise recommendations based on a larger pool of patients with EP–NENs and BMs.

## 5. Conclusions

Brain metastases in patients with EP-NENs are rare, with the greatest risk in those with grade 3 poorly differentiated NEC and have poor survival outcomes. Further collaborative studies into BMs in EP-NENs are needed to better our understanding of these patients.

**Author Contributions:** Conceptualization Z.-A.K. and M.G.M.; methodology, Z.-A.K., J.A., M.G.M.; validation Z.-A.K., X.W.; formal analysis, Z.-A.K., X.W., M.F.; investigation, Z.-A.K., J.A., M.D.; resources X.W., P.M., M.G.M.; data curation, Z.-A.K., J.A., M.G.M.; writing—original draft preparation, Z.-A.K., J.A.; writing—review and editing, Z.-A.K., J.A., M.D., M.F., B.C., P.M., C.M., W.M., A.L., R.H., J.W.V., M.G.M.; visualization, Z.-A.K.; supervision, M.G.M.; project administration, Z.-A.K., M.G.M. All authors have read and agreed to the published version of the manuscript.

**Funding:** This research received no external funding.

**Institutional Review Board Statement:** The study was conducted in accordance with the Declaration of Helsinki and approved by the internal review board of the Christie NHS Foundation Trust; the Clinical Audit Committee (reference 2746).

**Informed Consent Statement:** Patient consent was not applicable because this was a retrospective study and with no patient identifiers being used.

**Data Availability Statement:** Not applicable.

**Acknowledgments:** The salary for Zainul Kapacee was supported by the Christie Charity. Angela Lamarca is partly funded by the Christie Charity.

**Conflicts of Interest:** Z-A.K. received travel and educational support from EISAI. A.L. received travel and educational support from Ipsen, Pfizer, Bayer, AAA, SirtEx, Novartis, Mylan and Delcath; speaker honoraria from Merck, Pfizer, Ipsen, Incyte and AAA; and advisory honoraria from EISAI, Nutricia Ipsen, QED and Roche; she is also a member of the Knowledge Network and NETConnect Initiatives funded by Ipsen; all are outside of the scope of this work. R.H. has served on the advisory boards for Roche, BMS, Eisai, Celgene, Beigene, Ipsen and BTG. He has received speaker fees from Eisai, Ipsen, Mylan and PrimeOncology and has received travel and educational support from Bayer, BMS and Roche; all were outside of the scope of this work. J.W.V.: Consulting or advisory role for Agios, AstraZeneca, Delcath Systems, Keocyt, Genoscience Pharma, Incyte, Ipsen, Merck, Mundipharma EDO, Novartis, PCI Biotech, Pfizer, Pieris Pharmaceuticals, QED, and Wren Laboratories; Speakers' Bureau for Imaging Equipment Limited, Ipsen, Novartis, Nucana; received travel grants from Celgene and Nucana; all were outside of the scope of this work. M.G.M. received research grant support from Servier, Ipsen and NuCana. She has received travel and accommodation support from Bayer and Ipsen and speaker honoraria from Pfizer, Ipsen, NuCana and Mylan. She has served on advisory boards for Celgene, Ipsen, Sirtex, Baxalta and Incyte; all outside scope of this work. All other authors report no conflicts of interest.

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
