# Peer review of "The Management and Outcomes of Patients with Extra-Pulmonary Neuroendocrine Neoplasms and Brain Metastases"

_curroncol, doi:10.3390/curroncol29070405_

Round 1

Reviewer 1 Report

Dear authors,

thank you for sharing this interesting study on a topic under investigated.

Please find below my comments:

- please provide information about the volume of the brain metastases. You state that no patient was eligible for SRS, this should be demonstrated with data since literature shows new insight on this that should be considered in this paper (10.1016/j.ctro.2021.11.008)

- I would appreciate to have more information about the screening and imaging followup: did the authors performed 68Ga-DOTATOC assessment? Literature reports the utility of this methodology for meningiomas, neuroendocrine tumors and pituitary tumors. Since the interest on this is growing (10.3390/cancers14122925), it could be important to provide further information on this topic.

Author Response

Comment 1: Please provide information about the volume of the brain metastases. You state that no patient was eligible for SRS, this should be demonstrated with data since literature shows new insight on this that should be considered in this paper (10.1016/j.ctro.2021.11.008)

Response 1: Data on volume of brain metastases was obtained from the MRI or CT scan radiology reports. Data on size, location and number was intepreted from these written reports. At the Christie NHS Foundation Trust standard practice was for the patient’s case to be discussed at the Neuro-Oncology multidisciplinary team meeting (MDT) which involves the regional neurosurgical team, radiation oncology and radiologists. This point has been clarified in the results section lines 176-179. Factors such as disease status and patient performance status were considered and surgery or radiotherapy decisions decided at the MDT.

Recent ASTRO guidance http://www.ncbi.nlm.nih.gov/pubmed/34932393 excludes small cell lung cancer from their reccomendations for SRS owing to the paucity of data. As such we have only made suggestions that SRS may be useful in some cases. This is based on retrospectives studies of patients with predominantly pulmonary large cell carcinomas although these are likely to be biologically distinct from EP-NENs.

Comment 2 : I would appreciate to have more information about the screening and imaging followup: did the authors performed 68Ga-DOTATOC assessment? Literature reports the utility of this methodology for meningiomas, neuroendocrine tumors and pituitary tumors. Since the interest on this is growing (10.3390/cancers14122925), it could be important to provide further information on this topic.

Response 2: At The Christie NHS Foundation Trust, Gallium Dotatate scanning is used for baseline imaging for patients with advanced well-differentiated grade 1 and grade 2 gastroenteropancreatic neuroendocrine tumours to assess potential future suitability for treatment with Peptide Receptor Radionuclide Therapy (PRRT), or to assess disease status prior to consideration of potentially curative resection.  Gallium dotatate scanning was not employed post diagnosis of brain metastases.

Reviewer 2 Report

This is an interesting study on the analysis of neuroendocrine tumors affecting the central nervous system.

I think that both the abstract and the introduction make little focus on what is intended to be achieved and said by this study, the literature search does not seem to have been exhaustive, otre to ki-67 also the localization greatly affects the prognosis for these patients, not only for a question of access and surgical approach but also for histological features (see: Caporlingua A, Armocida D, Caporlingua F, Lapadula G, Elefante GM, Antonelli M, Salvati M. Disseminated Cerebrospinal Embryonal Tumor in the Adult. Case Rep Pathol. 2016;2016:6785459. doi: 10.1155/2016/6785459. Epub 2016 Oct 13. PMID: 27818821; PMCID: PMC5081462.).

It would be helpful to better describe the systemic treatment lines or make them more schematic so that they can be more easily referenced.

The discussion has greatly improved since the revision and is acceptable. 

The conclusions remain somewhat speculative and not very concise. One needs to summarize in extreme summary (2-3 sentences) the results and what new needs to be said.

Author Response

Thank you for your keen interest in the paper and raising very insightful points to improve the paper.

This is an interesting study on the analysis of neuroendocrine tumors affecting the central nervous system.

Comment 1: I think that both the abstract and the introduction make little focus on what is intended to be achieved and said by this study, the literature search does not seem to have been exhaustive, otre to ki-67 also the localization greatly affects the prognosis for these patients, not only for a question of access and surgical approach but also for histological features (see: Caporlingua A, Armocida D, Caporlingua F, Lapadula G, Elefante GM, Antonelli M, Salvati M. Disseminated Cerebrospinal Embryonal Tumor in the Adult. Case Rep Pathol. 2016;2016:6785459. doi: 10.1155/2016/6785459. Epub 2016 Oct 13. PMID: 27818821; PMCID: PMC5081462.).

 Response 1: Thank you, I have ammended the abstract and added a clear aim to the study within the background. Some of the results published have been re-arranged and simplified to deliver a clearer message.

I note your comments regarding the introduction and literature search. The focus was to provide a brief overview of EP-NENs to introduce the reader to the topic area and discuss the available literature on BMs in EP-NENs. There are only limited studies of a similar patient cohort. My concern with including studies of other histologically distinct disease sites in the introduction, such as embryonal tumours, was that its application to patients with EP-NENs and BMs becomes unclear. The final paragraph of the introduction has been ammended to make clearer the intentions of the study.

Comment 2: It would be helpful to better describe the systemic treatment lines or make them more schematic so that they can be more easily referenced.

Response 2: Thank you for this suggestion. The systemic treatments for gastroenteropancreatic NENs are wide ranging depending on histological grade, patient performance status and prior lines of therapy. Sequencing of therapies also varies, particularly for patients with grade 1 and 2 NETs.

Rather than describing the different treatments offered to patients which were previously listed in lines 141-142, I have referenced the comprehensive ESMO guidance “Gastroenteropancreatic neuroendocrine neoplasms: ESMO Clinical Practice Guidelines for diagnosis, treatment and follow-up”.

Comment 3: The discussion has greatly improved since the revision and is acceptable. 

 Response 3: Thank you.

Comment 4: The conclusions remain somewhat speculative and not very concise. One needs to summarize in extreme summary (2-3 sentences) the results and what new needs to be said.

Response 4: Thank you for this suggestion. The paragraphs within the conclusion have been shortened and added to complete the discussion. As suggested, a succinct 3 sentence conclusion has been added at the end.

Reviewer 3 Report

The manuscript has been improved, the revised version is now suitable for publication.

Author Response

Many thanks for your evaluation.

This manuscript is a resubmission of an earlier submission. The following is a list of the peer review reports and author responses from that submission.

Round 1

Reviewer 1 Report

The paper titled “Management and outcomes of patients with extra-pulmonary neuroendocrine neoplasms and brain metastases” from Kapacee and coworkers evaluates the incidence, clinico-pathological characteristics, presentation, management and survival outcomes of patients with EP-NENs and BMs at an ENETS CoE.

Although BMs are rarely reported in patients affected by NENs of non-lung origin, symptomatic BMs are associated with poor prognosis, so early detection and treatment could be advisable. Therefore, the topic is very interesting, adding an important and well-documented experience to the paucity of existing literature. Moreover, the authors, based on their experience, provide useful suggestions for clinical practice. The paper is well written and has a potential scientific interest, anyway some issues should be addressed.          

  1. In the introduction and discussion section the authors should consider a recent paper on metastatic patterns in NENs of different primary origin, in which BMs were reported only in patients with NEN of lung origin or unknown origin, but not in patients with gastroenteropancreatic origin (Hermans B.C.M doi.org/10.1159/000513249). Moreover, a recent analysis of SEER database showed that metastasis site plays an important role in survival of metastatic NEN patients independent of commonly described prognostic factors, please add some comments (Trikalinos N. A. 10.1186/s12902-020-0525-6).
  2. New recent literature data suggesting that NECs could be further subdivided into two prognostic distinct categories based on the Ki67 LI cut-off of 55%. The subgroup with a Ki67 LI < 55% has generally a better prognosis and should be treated and monitored differently from NECs with a Ki67 LI ≥ 55% (Feola T, Cancers 10.3390/cancers13061247). Therefore, the authors should add this consideration in the introduction section and if possible add new data on Ki-67 of NECs. Do you find any difference in BMs between NECs with lower and higher Ki-67 of 55%? Does the relative risk to develop BMs increase with the increase of Ki-67? If yes, the authors should consider this pathologic characteristic in their clinical suggestions.
  3. OS of patients with EP-NENs an BMs is influenced by grading and number of BM in univariate analysis, why the authors did not perform also a multivariate analysis to understand if these two prognostic factors are independent?
  4. In the discussion section lines 311-313 the authors affirms that upfront prophylactic cranial irradiation should be consider for patients with NECs. This sentence is not supported by evidence (no references) and I think it is “too strong” and not consistent with the followed sentences (323-327).
  5. Discussion line 342: considering the level of evidence, please replace “recommendations” with “suggestions”

Reviewer 2 Report

The topic is in my opinion very interesting analyzing the cases of BMs in patients with neuroendocrine disease. However, the way the study is presented in the abstract and in the introduction does not render well the study actually conducted. It is a retrospective analysis of 730 patients, but then in fact the cases with BMs are only 17 of which only 3 have undergone surgery, so it is very difficult to carry out an analysis of comparison between groups. Moreover, the basis on which the indication was given is not well clarified. It would be more favorable to carry out a descriptive analysis of an acceptable series of 17 patients. The discussion provided does not clarify the results and, in my opinion, was not sufficiently thorough. I would suggest to mention some clinical studies already performed on this topic and its variants (e.g. Caporlingua A, Armocida D, Caporlingua F, Lapadula G, Elefante GM, Antonelli M, Salvati M. Disseminated Cerebrospinal Embryonal Tumor in the Adult. Case Rep Pathol. 2016;2016:6785459. doi: 10.1155/2016/6785459. Epub 2016 Oct 13. PMID: 27818821; PMCID: PMC5081462.)

Reviewer 3 Report

Dear authors,

thank you for this interesting paper. Please find below my comments:

  • please define "localised radiotherapy" line 79 and in the text. Does systemic radiotherapy exists? Please explain this point and correct the text accordingly
  • line 187 "radical chemoradiotherapy" please consider to modify with definitive chemoradiotherapy
  • 333-359 could be considered a conclusion and I greatly appreciate it
  • line 285-287 you could also consider to cit 10.1016/j.ctro.2021.11.008
  • what about metabolic assessement with 
    68 Ga-DOTATOC PET/CT, that has been proven to be of benefit for the radiosurgical planning, during follow up showing SUV modification and total body detection of extraneural tumors not only for meningiomas but also for 

    neuroendocrine tumors and other tumors as well:

    • 10.3390/brainsci11030375; 
    • 10.3390/life11090942;
    • 10.2967/jnumed.119.241414
    • 10.1007/s00259-013-2659-5
    • 10.1007/s11307-014-0722-7.
    • 10.2967/jnumed.116.187138
